# SARS-CoV-2 multi-antigen protein microarray for detailed characterization of antibody responses in COVID-19 patients

**Alev Celikgil**[1]*, **Aldo B. Massimi**[1], **Antonio Nakouzi**[2,3], **Natalia G. Herrera**[1], **Nicholas C. Morano**[1], **James H. Lee**[1], **Hyun ah Yoon**[2], **Scott J. Garforth**[1], **Steven C. Almo**[1]*

**1** Department of Biochemistry, Albert Einstein College of Medicine, Bronx, New York, United States of America, **2** Department of Medicine, Division of Infectious Diseases, Albert Einstein College of Medicine and Montefiore Medical Center, Bronx, New York, United States of America, **3** Department of Microbiology and Immunology, Albert Einstein College of Medicine, Bronx, New York, United States of America

* steve.almo@einsteinmed.edu (SCA); acelikgi@einsteinmed.edu (AC)

## Abstract

Antibodies against severe acute respiratory syndrome coronavirus 2 (SARS-CoV-2) target multiple epitopes on different domains of the spike protein, and other SARS-CoV-2 proteins. We developed a SARS-CoV-2 multi-antigen protein microarray with the nucleocapsid, spike and its domains (S1, S2), and variants with single (D614G, E484K, N501Y) or double substitutions (N501Y/Deletion69/70), allowing a more detailed high-throughput analysis of the antibody repertoire following infection. The assay was demonstrated to be reliable and comparable to ELISA. We analyzed antibodies from 18 COVID-19 patients and 12 recovered convalescent donors. The S IgG level was higher than N IgG in most of the COVID-19 patients, and the receptor-binding domain of S1 showed high reactivity, but no antibodies were detected against the heptad repeat domain 2 of S2. Furthermore, antibodies were detected against S variants with single and double substitutions in COVID-19 patients who were infected with SARS-CoV-2 early in the pandemic. Here we demonstrated that the SARS-CoV-2 multi-antigen protein microarray is a powerful tool for detailed characterization of antibody responses, with potential utility in understanding the disease progress and assessing current vaccines and therapies against evolving SARS-CoV-2.

## 1. Introduction

Severe acute respiratory syndrome coronavirus 2 (SARS-CoV-2) has caused a pandemic with significant global impact since it was first identified in December 2019.

The immune responses of infected individuals have been studied to understand the pathogenesis of asymptomatic to severe disease using serological assays, primarily ELISA. Initially, the major antigen target for these assays was nucleocapsid (N) [1–4], one of the antigenic structural proteins; later, the focus has been on the spike (S), another antigenic structural protein [5, 6].

**Data Availability Statement:** All relevant data are within the paper and its Supporting information files.

**Funding:** This project has been supported in whole or in part by the Einstein Macromolecular

Therapeutics Development Facility, the Albert Einstein Cancer Center (P30CA013330) and the Price Family Foundation and contributions to the Albert Einstein Center for Experimental Therapeutics by Pamela and Edward S. Pantzer, the Wollowick Family Foundation Chair in Multiple Sclerosis and Immunology to S.C.A. The funders had no role in study design, data collection and analysis, decision to publish, or preparation of the manuscript.

**Competing interests:** The authors have declared that no competing interests exist.

The S glycoprotein consists of S1 and S2 domains; the S1 domain, which mediates binding to the host receptor angiotensin-converting enzyme 2 and the S2 domain, responsible for the fusion of host and viral cell membrane using six-helical bundle structure formed with heptad repeat (HR) domains [7]. Antibodies target multiple epitopes on the S protein and yet only some of them can induce neutralizing response against SARS CoV-2 [8, 9]. Specifically, neutralization has been shown to be proportional to the titer of antibodies that target receptor-binding domain (RBD) of S protein [10].

As SARS-CoV-2 evolves, new variants are emerging with mutations that potentially affect antigenicity. These variants have multiple mutations in RBD and non-RBD subdomains including D614G, E484K, N501Y, Deletion69/70, located in the S1 domain of S protein. Furthermore, some variants confer resistance to therapeutic monoclonal antibodies or to the antibodies elicited by vaccination [11]. While D614G and N501Y mutations showed minimal impact on neutralization, E484K mutation reduced neutralization by monoclonal antibodies, convalescent plasma therapy (CPT) and post-vaccination sera [12–15]. Since immunogenic epitopes are distributed across the entire S protein, more detailed characterization of antibody signatures is crucial not only for immune assessment, but also to evaluate CPT and vaccine responses for protection against infection with S variants.

Previously reported microarrays have been developed with only full-length S and N proteins or they included S protein domains expressed in non-mammalian systems without post-translational modifications (PTMs). S protein is heavily glycosylated in nature, and missing PTMs might fail the detection of antibodies to glycosylated determinants. During our manuscript preparation, we were not aware of any microarray studies combining full length S protein, its domains and variants to screen individuals for detailed analysis of antibody repertoire.

For this purpose, we developed a SARS-CoV-2 multi-antigen protein array with structural proteins N, S and its domains (S1, S2, RBD, HR2) and variants with single or double substitutions produced from mammalian cells. This assay allows parallel testing for antibody responses to different targets and domains of those targets using a very small sample volume. We were able to determine the antibody signatures of convalescent plasma (CP) donors who recovered from COVID-19, as well as multiple COVID-19 patients who received CP transfusion from these donors. Furthermore, we evaluated the binding of polyclonal antibodies in CP samples from individuals who were infected with SARS-CoV-2 early in the pandemic to S protein containing the clinically relevant substitutions D614G, E484K, N501Y and N501Y/Deletion69/70.

## 2. Materials and methods

### 2.1. Study design

We generated a SARS-CoV-2 protein microarray with two most antigenic structural proteins N and S along with S domains (S1, S2), RBD subdomain of S1, HR2 subdomain of S2, S variants with single (D614G, E484K, N501Y) and double (N501Y/Deletion69/70) substitutions (S1 Fig). After the quality of immobilized proteins were checked, the assay was validated with ELISA using eight CP donor plasma (Group 4, CONTAIN, Table 1). Multiple samples were screened in parallel on a single microarray slide designed with 12–16 subarrays. We analyzed the antigen-specific IgG, IgM and IgA responses in eighteen COVID-19 patients (Group 2, EAP_MMC, Table 1). Eleven of these patients were also evaluated for the antibody response to S variants. In addition, four COVID-19 patients were assessed before and after the CPT along with their donors. (4 from group 2 and group 3, EAP_MMC, Table 1).

**Table 1. Information on sera/plasma samples tested in this study.**

| Study group | N | Patient status | Characteristics | Source | Sample collection *data* |
|---|---|---|---|---|---|
| Group 1, Control | 3 | Not tested for SARS-CoV-2 | Pre-pandemic | NYBC | December 2019—February 2020 |
| Group 2, Patient | 18 | Severe and/or life threatening COVID-19 | Symptomatic for 3–7 days prior to transfusion | EAP_MMC Yoon et al. 2021 | April—May 2020 |
| Group 3, Donor | 4 | Recovered from COVID-19 | Asymptomatic for at least 14 days prior to sample collection | EAP_MMC Yoon et al. 2021 | March—April 2020 |
| Group 4, Donor | 8 | Recovered from COVID-19 | Asymptomatic for at least 14 days prior to sample collection | CONTAIN Ortigoza et al. 2022 | March—April 2020 |

Abbreviations: COVID-19, coronavirus disease 2019; EAP, expanded access protocol; MMC, Montefiore Medical Center; NYBC, New York Blood Center.

## 2.2. Sera/plasma samples

Control samples were obtained from healthy individuals at New York Blood Center (NYBC) between December 2019 and February 2020 (Group 1, Table 1). Samples were heat inactivated at 56˚C for 30 minutes and stored at 4˚C for short term or -80˚C for long term.

Remnant blood samples were available from patients who were positive by PCR for SARS-CoV-2 and received CPT as part of United States Expanded Access Program (EAP) [16] in April and May 2020 as described in Yoon et al. [17] (Group 2, Table 1). They were symptomatic for 3–7 days and hospitalized with severe COVID-19 for 3 days or less before the therapy. The remnant sera were obtained before and after transfusion (day 0, 1, 3, 7). CP donor samples were collected from individuals who were positive by PCR for SARS-CoV-2 and symptom free for at least 14 days before the plasma collection as part of an institutional donor program conducted in March-April 2020 as described previously [16] and used in EAP or CONTAIN COVID-19 trial [18] (Groups 3, and 4, Table 1).

The retrospective cohort study, the donor plasma procurement protocol, and the use of the EAP were approved by the Albert Einstein College of Medicine (AECOM) IRB. The retrospective cohort study that included collection of remnant blood samples of patients was approved by the AECOM IRB for human subjects with a waiver of informed consent.

## 2.3. Plasmids

Ectodomain of SARS-CoV-2 S (residues 1–1208) in pCAGGS vector was kindly provided by McLellan and coworkers [19]. S2 domain (residues 687–1208) in pCAGGS vector was kindly provided by John Lai (Einstein). S1 domain (residues 13–685), RBD of S1 (residues 319–541) and HR2 of S2 (residues 1163–1202) were amplified by polymerase chain reaction (PCR) using SARS-CoV-2 S as template and subcloned into pcDNA3.3 vector with a C-terminal hexahistidine tag using In-Fusion cloning technology (Takara Bio USA, Inc.). RBD of SARS-CoV S was cloned into pcDNA3.3 vector from pcDNA3.1-SARS-Spike (Addgene plasmid#145031). SARS-CoV-2 N was cloned into a pSGC-his vector as described in our previous paper [20]. Single and double mutations were introduced using PCR and In-Fusion with the McLellan S expression construct.

## 2.4. Antigen expression and purification

S and N proteins were expressed and purified as previously described in Herrera et al. [20]. S1, S2, HR2 and S variants were expressed in ExpiCHO-S™ cells as described for full-length S. RBD of SARS-CoV2 and SARS-CoV S were transiently expressed in FreeStyle™ 293-F cells (Thermo Fisher Scientific). Cells were resuspended at $1 \times 10^6$ per mL with fresh media on the

day of transfection. DNA (0.5mg/ml) and Polyethylenimine (2mg/ml; PEI, Polysciences, Inc., 23966) were mixed in 1XPBS and incubated for 15 minutes at room temperature. The DNA/ PEI mixture was added to the cells dropwise and cells were incubated in a shaker at 37˚C and 5% CO2 [21]. 24 hours post-transfection, valproic acid salt (Sigma-Aldrich, P4543-100G) suspended in media (1/3 of starting volume) was added to the cells at final concentration of 3mM [22]. The cells were harvested on day 7 post-transfection. S domains and variants were purified similar to S protein using nickel resin (10 mL/L, His60 Ni2+ superflow resin, Takara cat# 635664) with wash buffer (25 mM MES, 150 mM NaCl, 10% glycerol, 50 mM Arg-Cl, 5mM imidazole, pH 6.5) and elution buffer (25 mM MES, 150 mM NaCl, 10% glycerol, 100 mM Arg-Cl, 0.5 M imidazole, pH 6.50). The eluates were concentrated using Amicon centrifugal units (EMD Millipore) and dialyzed against 50 mM Tris, 250 mM NaCl, pH 8.0 for 2 hours at room temperature and then overnight at 4˚C. Proteins were analyzed by SDS-PAGE. Protein concentrations were determined using UV absorbance at 280nm; extinction coefficient was calculated from amino acid sequence using Expasy online ProtParam.

## 2.5. Multi-antigen protein array production and processing

A protein array of SARS-CoV-2 antigens was generated with the full-length N and S, S1, S2, RBD and HR2 domains of S protein. Negative controls included 1XPBS, sample buffer and human acetylcholinesterase (huAche). RBD of SARS-CoV S was included for cross-reactivity. Human immunoglobulin (Ig) isotype controls IgG, IgM and IgA were printed as reference to normalization for each sample, and to confirm detection by the secondary antibodies.

Each slide was printed with twelve identical subarrays, and used for serum dilutions or multiple serum screening. Target antigens were adjusted to between 0.3–7.2 femtomol (fmol), and Ig isotype controls 0.15–3.6 fmol per spot. Microarray slides were generated with aminosilane-coated slides using a piezoelectric printer (Arrayjet, Edinburgh, UK) and processed as described previously [20]. To determine levels of immobilized proteins, one slide was probed with Alexa Fluor® 647 anti-his antibody (1:250, BioLegend, 362611). Sera/plasma samples were single or serial diluted (three-fold dilutions from 1:90) in 250 μl 5% milk, PBS, 0.2% Tween-20.

## 2.6. Data analysis

Data was processed using GenePix® Pro7 software (Molecular Devices). Each sample was corrected for background by subtracting the raw spot intensity of negative control protein from every sample spot. The results were normalized relative to corresponding concentration of Ig isotype controls in each subarray to minimize the variations that occur during sera and array processing. The mean fluorescence intensity ± standard deviation (SD) was calculated from the replicates for each antigen concentration.

Results were plotted using GraphPad Prism software version 8.4.3. Nonlinear regression model was used for dilution-response curves of serum samples. The bar plots and heatmap were used to display antibody levels of multiple samples against antigen targets, CPT time points or Ig isotypes. The data was also represented as the ratio of antigen specific antibody levels.

## 2.7. ELISA

SARS-CoV-2 N, S2, RBD, and HR2 protein-binding IgG were measured by ELISA using donor CP as previously described in Bortz et al. [23].

## 3. Results

### 3.1. Generation and validation of SARS-CoV-2 multi-antigen protein microarray

We initially generated a protein array of SARS-CoV-2 antigens with structural proteins N, S, S domains (S1, S2) and subdomains (RBD, HR2). RBD domain of SARS-CoV was also included for its cross-reactivity with SARS-CoV-2. Later, we extended the array to S variants with single (D614G, E484K, N501Y) and double mutations (N501Y/Deletion69/70). While SARS-CoV-2 N protein was produced from *E. coli* cells, we used mammalian cells to express SARS-CoV-2 S protein, its domains and variants with PTMs. Representative SDS-PAGE analysis of proteins is shown in Fig 1A.

After a concentration series of antigens, Ig isotype controls (IgG, IgM and IgA) and negative controls were arrayed onto the microarray slide (Fig 1B and 1C), the levels of immobilized proteins were determined using antibodies against the his tag (Fig 1D). A pattern of decreasing signal intensities correlated to antigen amount was observed, and no signal was detected from buffer spots. Multiple samples were screened in parallel on a single microarray slide designed with twelve subarrays. To visualize and analyze the human antibodies bound to the antigens on the slide, fluorescent-labeled secondary anti-human antibodies were used. Representative images are shown as two-color images (IgG/IgA) for serum antibody detection of a COVID-19 patient one day before (Day -1) and one day after (Day 1) convalescent plasma transfusion using secondary antibodies for IgG (red) and IgA (blue) in Fig 1D.

### 3.2. Assay development

To identify a serum dilution that gives reliable detection for multiple samples with different antibody titers, 11 dilutions of a single sample were tested for IgG response to S and N proteins (Fig 2A and 2B). Next, 21 samples (Group 1 and 2, Table 1) were assayed at the 3 highest dilutions (1/90, 1/270, 1/810) (Fig 2C). At all three dilutions, antibody response to S and N proteins could be detected in the sera/plasma; antibodies were not detected in samples from control subjects. 1/270 dilution was chosen for further analysis of multiple samples to avoid the false negative results observed with low titer samples at higher dilutions.

### 3.3. Protein array and ELISA comparison

We also compared our assay to ELISA which requires multiple plates to screen samples against different targets. When the same target antigens and same donor plasma samples (Group 4, Table 1) were used, results from protein microarray assay with 1/270 serum dilution and 2.4 fmol of each target antigen were comparable to ELISA which uses antigen concentration at saturation level (S2 Fig). Both assays identified the same samples with high levels of IgG for N, S2 and RBD, and also that they were seronegative for HR2 subdomain of S2.

### 3.4. Antibody signatures of COVID-19 patients

Antibody responses of eighteen COVID-19 patients who were infected with SARS-CoV-2 between April and May 2020 were analyzed, in parallel with three control plasma samples of unknown infection history that were obtained between December 2019 and February 2020 (Group 1 and 2, Table 1). Multi-antigen screening showed higher IgG for S than N in 14 of the COVID-19 patients (Fig 3). Qualitative analysis of antibody responses to S domains (S1, S2) and subdomains (RBD, HR2) showed higher S2 IgG than S1 and RBD IgG in most of the samples (16 and 11 samples, respectively). IgM and IgA reactivity in the patient sera were overall less than IgG (S3A Fig). S1/RBD IgG ratio was low in all of the samples, showing antibody

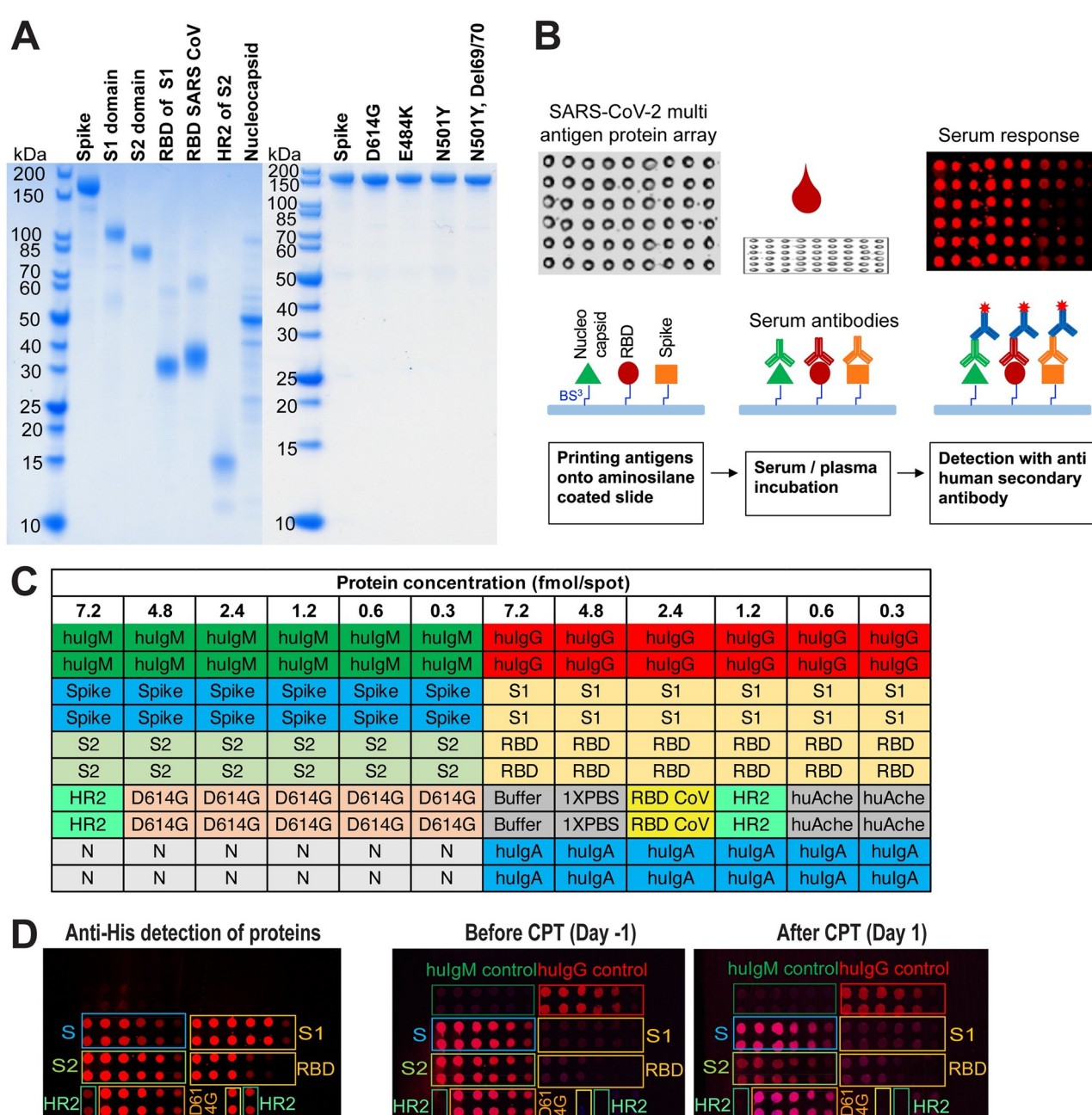

**Fig 1. SARS-CoV-2 multi-antigen protein array production and processing.** (A) Purified array proteins analyzed by SDS-PAGE under reducing conditions. (B) Schematics of workflow. (C) A representative protein array template showing SARS-CoV-2 antigen target proteins, human immunoglobulin controls (huIgM, huIgG, huIgA), buffer and negative controls along with the protein amounts and duplicates of each sample. (D) Anti-his staining of antigen target proteins immobilized on the microarray slide, and serum antibody detection of a COVID-19 patient one day before (Day -1) and one day after (Day 1) convalescent plasma transfusion using secondary antibodies for IgG (red) and IgA (blue) (two-color image). huIg control proteins do not have his tags; N, nucleocapsid; S, spike protein.

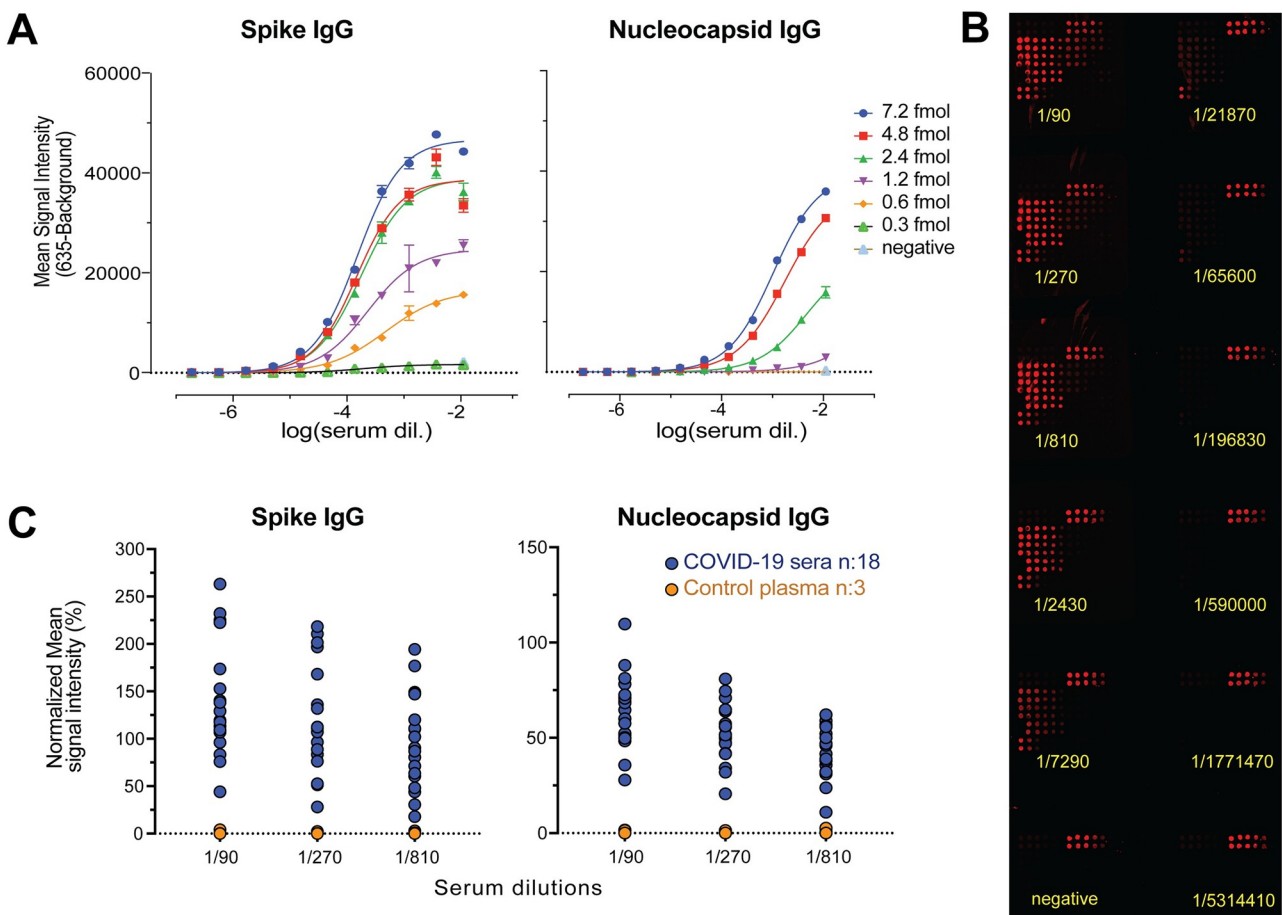

**Fig 2. Serial dilution test to identify a single dilution that gives reliable detection for multiple sera/plasma samples.** (A) Spike and nucleocapsid IgG levels are shown for single serum sample at 11 serial dilutions against six protein concentrations. Mean values ± SD are shown for duplicates. (B) Representative image of protein microarray printed with 12 identical subarrays that was used for IgG detection in a single sample with three-fold serial dilutions. (C) Spike and nucleocapsid IgG levels are shown for multiple samples (n:21) with three highest serial dilutions. The results for five antigen concentrations (7.2, 4.8, 2.4, 1.2, 0.6 fmol) are normalized relative to corresponding concentration of IgG isotype controls, and averaged. The mean values are calculated for the replicates of target antigen for each sample.

responses to S1 domain was mostly targeting RBD (S3B Fig). We observed a wide range of S/RBD and S2/RBD ratios between individuals indicating different levels of serum antibodies targeting non-RBD regions on S2 compare to RBD.

There was no antibody response to the RBD domain of SARS-CoV in any of the serum samples in our screening (representative image is shown in Fig 1D).

### 3.5. Antibody signatures of CPT donors and recipients

Antibody signature of CP has been proposed to affect the efficacy of CPT [24, 25]. We screened samples from four COVID-19 patients and their donors who recovered from COVID-19 (Group 2 and 3, Table 1) against SARS-CoV-2 S, S2, RBD, HR2 and N before and after CP transfusion. Two of the CPT recipients died 3 days (patient#1) and 18 days (patient#2) post-transfusion; two recovered 52 days (patient#3) and 2 days (patient#4) post-transfusion. While we observed differences in antigen specific antibody responses between patients and donors, we were also able to monitor the changes in antibody levels at different time points post-CPT

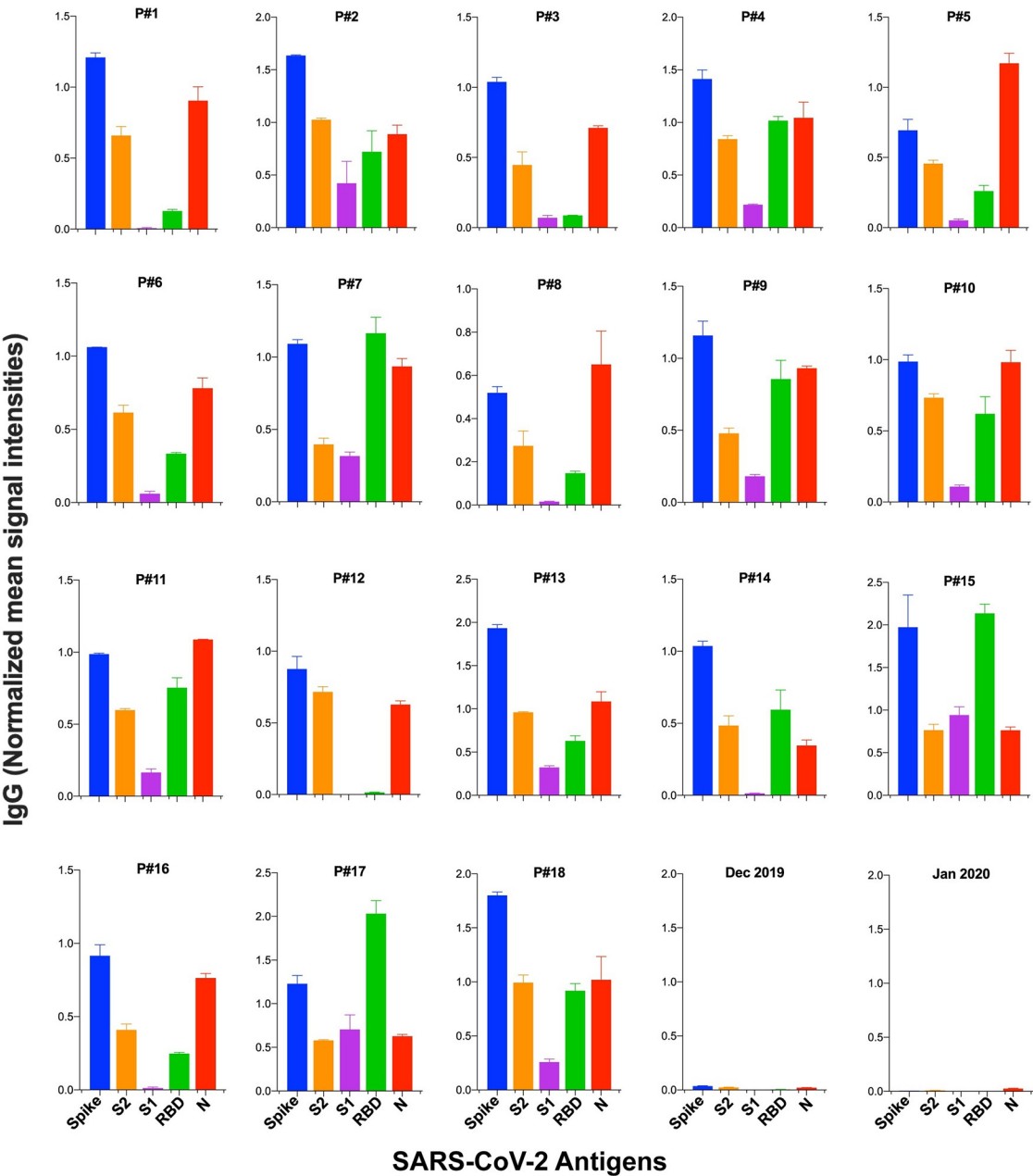

**Fig 3. Serum IgG levels against SARS-CoV-2 antigens in COVID-19 patients.** Serum IgG levels are shown against spike, its domains (S1, S2, RBD) and nucleocapsid (N) for 18 patients sera (P#1–18) and 2 control plasma samples. Signal intensities are normalized to human IgG isotype control for each sera/plasma and mean values ± SD are calculated for duplicates. The data is shown for a single dilution (1/270) and 7.2 fmol antigen concentration.

(S4 Fig). Although all the patient samples were seropositive for S2 IgG, we did not observe antibody response to HR2 subdomain of S2 in any of the samples.

### 3.6. Antibody responses to spike variants

In order to begin to address whether patient's antibodies against SARS-CoV-2 from early pandemic could show response to later isolates, we extended our protein microarray panel to S

variants with single mutations D614G (in CTD), E484K (in RBD), N501Y (in RBD) and double mutations N501Y/Del69/70 (in RBD/NTD) in S1 domain (S5 Fig). We screened eleven COVID-19 patients along with two control samples (Group 1 and 2, Table 1). While we observed serum responses to S variants with single and double mutations in all of the patient samples tested (Fig 4A and 4B), the IgG bound to D614G variant was ~2-fold lower than the wildtype S-specific IgG (Fig 4C). IgG bound to E484K, N501Y, and N501Y/Del69-70 variants were comparable to wild-type S-specific IgG levels in most of the samples. Only one patient had slightly higher antibody response to these three variants than wildtype S (patient#2). In three patients (P#7, P#9, P#11), IgG response to E484K, which is in RBD domain, was less than the response to the other RBD mutation N501Y (Fig 4B). IgM levels were uniformly lower than IgG, which could be due to the time interval between symptom onset and the sample being taken. Although the patients 1, 6, 8 and 11 had higher levels of S-specific IgM compared to other patients, they were still less than IgG. In these samples, the differences between E484K and N501Y or N501Y/Del69/70 were higher for IgM than IgG, however the antibody levels were low.

## 4. Discussion

For a detailed characterization of antibody responses against SARS-CoV-2 antigens, we developed a multi-antigen protein array initially with the two of the most immunogenic SARS-CoV-2 antigens S and N. Since antibodies target multiple epitopes on the S protein, and the antigenicity of these epitopes change due to mutagenesis in emerging SARS-CoV-2 variants [11], we extended the array to S domains and variants with single or double substitutions. RBD of SARS-CoV was also included in the array for cross-reactivity due to high protein similarity between SARS-CoV and SAR-CoV-2. S protein is heavily glycosylated, and its expression in non-mammalian systems might fail the detection of antibodies to glycosylated determinants, therefore we produced the proteins from mammalian cells for protein microarray fabrication [19, 26]. We also demonstrated that protein microarrays are comparable to ELISAs, which have been used widely for serological testing during the pandemic. Both assays not only indicated that same samples are seropositive or seronegative, but also showed similarity for the sample groups with high-medium-low levels of antibodies. The difference in response seen between the protein array and the ELISA is simply because of the high fixed concentration of immobilized protein utilized in the ELISA. This could increase sensitivity of the ELISA relative to the protein array, particularly for low affinity antibodies, but may reduce the linearity of the response when comparing antigens that are at opposite ends of the detection spectrum.

In our serum screening, all of the individuals who were infected with SARS-CoV-2 had reactivity with the full-length S protein, but binding to distinct S domains differed. Furthermore, we showed that sera from early in the pandemic contained antibodies capable of binding SARS-CoV-2 S protein containing the clinically relevant substitutions D614G, E484K, N501Y and N501Y/Deletion69/70, prevalent in clinical isolates from later in the outbreak.

In this study, our screening of COVID-19 patients showed high S-specific IgG levels in comparison to N-specific IgG in most of the patients as shown in previous studies [25, 27]. We also studied the immunogenic features of specific subdomains; RBD in the S1 domain and HR2 in the S2 domain of S. The RBD subdomain of S1 is a highly variable domain of S protein, a low ratio of S1 to RBD IgG observed in patients sera was suggesting that RBD is the antigenic determinant of S1. The HR2 subdomain of S2 plays a major role in the viral fusion to the cell membrane, and is a target of fusion inhibitors against SARS-CoV-2 [28]. We did not observe any signal against HR2 in samples, even those with high levels of S2-specific antibodies. The wide range of S/RBD and S2/RBD IgG ratios in samples suggest a different distribution of dominant epitopes in the S domains between individuals. Although RBD is the main antigenic

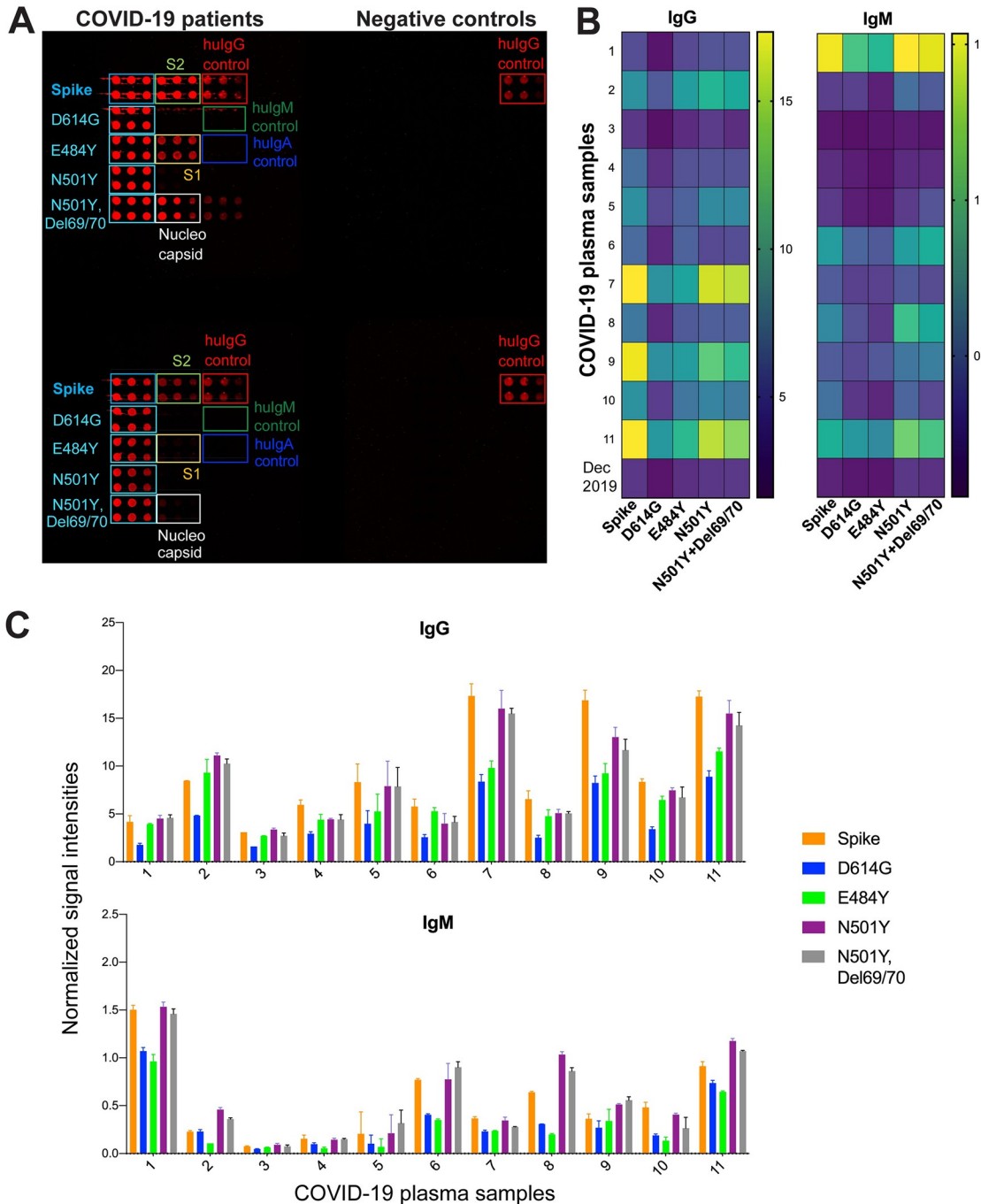

**Fig 4. Antibody responses to S variants with single and double mutations in sera from COVID-19 patients from the beginning of the pandemic.** (A) Representative image of four subarrays from SARS-CoV-2 protein microarray slide used for screening of two COVID-19 patients' sera and two control plasma samples. Proteins are printed as 4.8–2.4–1.5 fmol in duplicates. IgG (red) responses are shown against S variants with single mutation D614G, E484K, N501Y and double mutations N501Y with Del69/70 along with S, S1, S2 and N. (B) Heatmap showing IgG and IgM profiles of patients' sera (n:11) and control plasma (Dec 2019) against 2.4 fmol of each S variant. Signal intensities from duplicates are averaged. (C) Antibody levels are shown as bar plots and data are represented as normalized mean signal intensities ± SD.

region in the S1 domain, most of the COVID-19 patients showed higher levels of antibody binding with non-RBD targets in the S2 domain, as reported in data from other studies [8, 29]. Also the S2 antibody signature between patients was similar to the full-length S protein.

SARS-Cov-2 S protein shares the highest protein similarity with SARS-CoV S protein compared to other human coronaviruses. Furthermore, Du and colleagues who developed a protein array using only S and N proteins from SARS-CoV-2 and other human coronaviruses showed that antibodies against SARS-CoV-2 N protein cross-reacts only with SARS-CoV N protein [30]. While all the individuals we have tested in this study were positive for SARS-CoV-2 N protein, they did not show any antibody response to RBD domain of SARS-CoV.

IgM and IgA antibodies were low in most of the patient samples; the presence or absence of which may reflect differences in time of exposure to the virus and the individual's immune responses. While the presence of IgM is an indication of early disease, IgA is detected in more severe disease outcomes [31–33]. Target and isotype differences of serum antibodies is important for the immune assesment of individuals infected with SARS-CoV-2. Furthermore, immunoprofiling using multi-antigen protein arrays with only a couple of microliters of sera could help to detect distinct antibody characteristics of CPT donors and recipients, and monitor the disease progress post-therapy. An advantage of the protein chip for detecting antibodies is the ability to assay many samples, in this case time points, in parallel. Whilst it is clear that the patients tested showed different levels of antibodies against the antigens tested (S4 Fig), the sample size would have to be greatly expanded to detect a possible correlation with CP therapy.

Although SARS-CoV-2 protein microarrays were used for serum screening of CP in previous studies [34, 35], S variants were not included in the arrays, and S2 was not expressed in mammalian systems which could affect antigenicity due to missing PTMs and therefore cannot be used to compare impartially to different domains that were expressed in a mammalian system.

As new SARS-CoV-2 variants are emerging, reinfection cases are increasing all around the world. These variants have several mutations located in the subdomains of S protein (S5 Fig). From a few cases studied, Tillett et al. reported an asymptomatic reinfection case which was infected with two SARS-CoV-2 variants that both had single D614G substitution in S and several different mutations outside of S protein [36]. Prado-Vivar et al. reported a reinfection case with worse disease than the first infection [37]. This individual was infected with S variant with single D614G substitution at first infection, and wild type S at second infection. These individuals were infected with variants that had also several different mutations outside of S protein. There was no data on these individuals' antibody responses to S and its domains. In our screening of individuals infected with SARS-CoV-2 early in the pandemic, their response was mounted against the virus they were infected with, and their antibodies bound to other variants with single mutations D614G, E484K, N501Y and double mutations N501Y/Del69/70 in the S1 domain of S. This is likely due to polyclonality of antibodies in the sera/plasma, targeting a broad range of epitopes, as previously suggested [14]. However, specific combinations of mutations in S protein, as well as in other immunogenic viral proteins could diminish the protective immunity from previous infection and increase the probability of reinfection. SARS-CoV-2 Beta (B.1.351) variant, which includes E484K, D614G, N501Y and other mutations in S, exhibited a reduced neutralization by plasma from individuals infected with SARS-CoV-2 early in the pandemic or vaccinated [11]. All of the samples we tested showed some reduction in response to S with a single D614G substitution, but only a few samples to E484K; this presumably reflects the differences between individuals' antibody repertoire.

Currently, there are different types of vaccines against SARS-CoV-2 such as mRNA, viral vector and protein subunit as well as traditional inactivated virus [38]. While inactivated whole virus delivers native immunogenic epitopes to induce immune response in the host, the others use an immunodominant protein subunit, typically spike protein, as the target. To

increase efficacy of these vaccines against SARS-CoV-2 variants, different strategies have been tested including administration of multiple vaccine doses and using spike protein with substitutions to match the latest variant in multiple subdomains as target antigen. Despite all these efforts, immunization rates are low in some populations due to limited supply or vaccine hesitancy. Consequently, protection against SARS-CoV-2 is poor resulting in locally evolving variants that impact the effectiveness of treatments. Kunze and colleagues showed that locally sourced convalescent plasma used for CPT in COVID-19 patients was associated with lower mortality than distantly sourced convalescent plasma, which suggested the immune responses show geographical differences [39]. As the humoral immune response between individuals becomes more heterogeneous due to emerging variants, different types of vaccines and vaccination rates, serological assays which only use the entire S and N proteins as antigen targets would not be sufficient to study the antibody repertoire of infected or vaccinated people. Protein microarrays developed with broad antigen targets such as N and S protein with its domains/subdomains, variants, and other human coronavirus antigens (cross reactivity) would be an ideal tool for determining detailed antibody signatures. Correlation, or otherwise, of these antibody signatures with clinical outcome would allow for optimization of future vaccines, and identification of potential leads for monoclonal antibody treatment.

Although we used the SARS-CoV-2 protein array for detailed characterization of individuals' immune responses, this study is limited to showing the antibody repertoire but not the efficacy of the antibody response in controlling infection. The protein array does not assess neutralizing activity, nor does it measure antibody avidity.

The strength of our study was screening each individual against multiple targets and concentrations in the same subarray with single small sample volume unlike ELISAs which requires multiple plates and sample aliquotes. Moreover, this assay could be used in a point-of-care setting to provide individuals' full antibody signatures by targeting multiple antigens unlike most of the current point-of-care tests which target only a single antigen.

Most importantly, our protein microarray, allowing parallel analysis of multiple variant proteins and their subdomains, is ideally suited for the analysis of the potential of patient antibodies to neutralize variants other than that which was responsible for the initial infection. Further, it is hoped that by extending the scope of the study and analyzing the full spectrum of antibody responses, we can determine immunogenic antigens that are less prone to loss through viral evolution.

## Supporting information

**S1 Fig. Study design for SARS-CoV-2 multi-antigen protein array.**
(PDF)

**S2 Fig. IgG responses to nucleocapsid protein (N) and S domains detected with protein microarray and ELISA.** Sera with highest antibody responses are highlighted with blue frame for each target antigen for both assays. (A) IgG responses to N, S2 domain, RBD and HR2 subdomains determined from protein array for eight convalescent sera along with negative and positive controls. Positive control is a convalescent plasma sample with high S antibody levels determined with ELISA. Signal intensities from duplicates of each target antigen are averaged and normalized to corresponding concentration of human IgG isotype control. (B) IgG responses to N, S2, RBD and HR2 in the same eight sera are detected with ELISA in eleven three-fold serial dilutions starting from 1/100. Absorbance at 450nm is averaged for duplicates and represented using non-linear regression model.
(PDF)

**S3 Fig. Immunoprofiling COVID-19 patients against multiple SARS-CoV-2 antigens.** (A) Bar plots are showing IgG, IgM, and IgA response to 4.8 fmol of each S, S2, S1, RBD and N antigen for 18 patients who were infected early in the pandemic. The signal intensities are normalized relative to corresponding concentration of Ig isotype controls and mean values are calculated for the replicates of target antigen for each plasma sample. (B) The ratios of S, S2 and S1 to RBD domain are shown as bar plots for IgG.
(PDF)

**S4 Fig. IgG profiles of four CPT donors and four recipients against SARS-CoV-2 proteins before and after CPT.** IgG serum responses are shown as percentages of normalized mean signal intensities against 4.8 fmol antigen proteins spike, S2, RBD, HR2 and nucleocapsid. CPT time points for the recipients are shown as one day before the transfusion, one and three days after the transfusion along with corresponding donor's antibody levels (CP).
(PDF)

**S5 Fig. Shared spike mutations in SARS-CoV-2 variants.** S variants' shared mutations E484K, N501Y, D614G and Deletion69/70 are shown with their location on S1 domain of S protein. NTD, N-terminal domain; RBD, Receptor binding domain; CTD, C-terminal domain; FP, Fusion peptide; HR1, Heptad repeat 1; HR2, Heptad repeat 2.
(PDF)

**S1 Raw images.**
(PDF)

# Acknowledgments

We thank Liise-anne Pirofski for her insightful discussions and editorial input for the manuscript. We also thank Jason S. Mclellan for providing spike plasmid, and Jonathan R. Lai for sharing S2 plasmid.

# Author Contributions

**Conceptualization:** Alev Celikgil, Scott J. Garforth, Steven C. Almo.

**Data curation:** Alev Celikgil.

**Formal analysis:** Alev Celikgil, Aldo B. Massimi.

**Funding acquisition:** Steven C. Almo.

**Investigation:** Alev Celikgil, Antonio Nakouzi, Natalia G. Herrera, Nicholas C. Morano, James H. Lee, Hyun ah Yoon.

**Methodology:** Alev Celikgil, Aldo B. Massimi, Scott J. Garforth, Steven C. Almo.

**Resources:** Antonio Nakouzi, Natalia G. Herrera, Nicholas C. Morano, James H. Lee, Hyun ah Yoon.

**Supervision:** Scott J. Garforth, Steven C. Almo.

**Validation:** Alev Celikgil.

**Visualization:** Alev Celikgil.

**Writing – original draft:** Alev Celikgil.

**Writing – review & editing:** Alev Celikgil, Scott J. Garforth.

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
