## [Decision Letter · Decision Letter 0]

14 Nov 2022

PONE-D-22-28228SARS-CoV-2 multi-antigen protein microarray for detailed characterization of antibody responses in COVID-19 patientsPLOS ONE

Dear Dr. Celikgil,

Thank you for submitting your manuscript to PLOS ONE. After careful consideration, we feel that it has merit but does not fully meet PLOS ONE’s publication criteria as it currently stands. Therefore, we invite you to submit a revised version of the manuscript that addresses the points raised during the review process.

1) The authors should the discussion with the implications of the technology used and the results obtained in light of the complex scenario the world is facing (different VOCs, immunization rates depending on the region, different vaccines, prime and booster doses, ages, clinical outcomes and much more);2) Please, answer all the questions raised by the both reviewers.

We look forward to receiving your revised manuscript.

Kind regards,

Paulo Lee Ho, Ph.D.

Academic Editor

PLOS ONE

Journal Requirements:

3. Please upload a copy of Figure 5, to which you refer in your text on page 17. If the figure is no longer to be included as part of the submission please remove all reference to it within the text.

Reviewers' comments:

Reviewer's Responses to Questions

**Comments to the Author**

1. Is the manuscript technically sound, and do the data support the conclusions?

Reviewer #1: Yes

Reviewer #2: Partly

2. Has the statistical analysis been performed appropriately and rigorously? 

Reviewer #1: Yes

Reviewer #2: N/A

3. Have the authors made all data underlying the findings in their manuscript fully available?

Reviewer #1: Yes

Reviewer #2: Yes

4. Is the manuscript presented in an intelligible fashion and written in standard English?

Reviewer #1: Yes

Reviewer #2: Yes

5. Review Comments to the Author

Reviewer #1: The manuscript "SARS-CoV-2 multi-antigen protein microarray for detailed characterization of antibody" by Alev Celikgil et al. introduces a valuable technique to screen individuals against multiple targets of SARS-CoV-2 antigens and variants in the same subarray with single small volume. The authors improved previously reported microarrays by using mammalian cell-derived antigens and showed the possibility of understanding the disease progress, detecting the efficacy of convalescent plasma transfusing therapy, or applying this technique to point-of-care testing.

Although this technique is quite interesting, some parts of this manuscript need to be explained for readers. Therefore, this reviewer would like to suggest some parts that should be clarified.

Major points:

1) The authors are encouraged to clearly state the rationale for developing this technique. As the authors mentioned in the Discussion, other microarrays against SARS-CoV-2 are already available. The advantageous ore incremental points for developing this technique should be explained in the Introduction compared to the others.

2) The authors focused on the only 4 variants D614G, E484K, N501Y, and N501Y/Deletion69/70. How were these VOC selected with respect to any clinical relevance of these variants?

3) In Fig. 2A, according to the fitting curves of serum signal intensity against 6 concentrations of Spike antigen, the curve in 2.4 femto-mol was not likely to show linearity. Although the authors mentioned their protein array was compared with an ELISA at lines 232-239 of page 11 and Fig. S2, the authors are encouraged to show in detail the detection capacity of this technique compared to a conventional ELISA.

4) Table 1 indicates the information on blood samples the authors tested in this study. However, this reviewer could not follow which samples were used for which test sufficiently. For example, the authors mentioned, "Eleven of these patients were also evaluated for the antibody responses to S variants" in lines 88-89 of pages 4-5. Still, its result in Fig. 4B showed "Convalescent plasma samples" at the Y axis. This expression seemed to lead to misunderstanding whether the sample is from patients or donors.

5) On page 17, lines 354-356, it states, “Target and isotype differences of serum antibodies combined with clinical features may be useful predictors of disease progress for individuals infected with SARS-CoV-2." Even though the samples used in this study include a wide range of disease states, from severe to asymptomatic, no predictive analysis was performed. The authors should only mention this if there is confirmation of an ability to predict an individual outcome with findings using this technique.

6) Functional aspects of antibodies that contribute to protection from infection include not only antibody titer but also neutralizing activity and avidity. Of course, the microarray is intended to detect antibody titers. Therefore, the authors are required to explain the limitations of this technique.

Minor points:

1) The authors suddenly used the abbreviation "CPT" on line 128 of page 6 and line 170 of page 8. It may indicate "Convalescent Plasma Transfusing" or "Convalescent Plasma Therapy," the author should clarify the abbreviation.

2) "Fig. 5" at line 362 of page 17 may be incorrect. It seems "Fig. S4".

Reviewer #2: The present study aims to design and produce immunogenic proteins of SARS-CoV2 (S and N) and different variants of S proteins and evaluate reactivity of serum sample of infected patients on a multi-antigen protein microarray.

Major issues:

-How such microarray could help as a POC or predict the outcome of the illness in patients according to the fact that the humoral immune response is very heterogenous in patients (as authors and previous similar works did not get any rational correlation between pattern of humoral immune responses and outcome of the disease). Moreover, the heterogeneity of humoral immune response become more and more complex as people infected with new emerging variants and more than 80% of them have vaccinated with different kinds vaccines eg. Inactivated whole virus, S, S1 and RBD subunit vaccine.

-How is it possible after normalization, specific IgG against RBD is higher than specific IgG against S1 in keeping with the truth that the molar ratio of coating RBD and S1 recombinant protein are equal. In that case, S1 should have more binding epitopes than RBD!

-Although the SDS-PAGE and anti-His tag antibody indicating presence of the HR domain, there is not any sign of reactivity of serum samples with this region, even positive control is not working!

-The reactivity pattern of serum belongs to the 12 recovered convalescent donors, which mentioned in abstract and table1, was to properly determined in the MS.

-Different studies have shown that there are cross-reactive antibodies against N and S proteins in serum patient with other coronavirus family which could affect the interpretation of the results, why the authors have not used such controls in their experiments.

Minor Issues:

-Abbreviation of “huAche” should be added at page 7 line 148

-Using day-1 and day1 in figure1 and description in result section is unclear.

- The origin of expressed N protein at page 9 line 186 is unclear.

- The concentration of antigens has not mentioned in result section at page 11 lines 226-230.

- Why positive control in S2 figure has not any reactivity with HR antigen.

- Where is the figure 5 which has been mentioned at page 17 line 362!

6. PLOS authors have the option to publish the peer review history of their article (what does this mean?). If published, this will include your full peer review and any attached files.

Reviewer #1: No

Reviewer #2: No

---

## [Author Response · Author response to Decision Letter 0]

24 Dec 2022

Our responses to the reviewers’ comments are as follows: 

1) The authors should the discussion with the implications of the technology used and the results obtained in light of the complex scenario the world is facing (different VOCs, immunization rates depending on the region, different vaccines, prime and booster doses, ages, clinical outcomes and much more);

We have added the following paragraphs to the discussion on pages 19-21 to address this important point: 

‘Currently, there are different types of vaccines against SARS-CoV-2 such as mRNA, viral vector and protein subunit as well as traditional inactivated virus [38]. While inactivated whole virus delivers native immunogenic epitopes to induce immune response in the host, the others use an immunodominant protein subunit, typically spike protein, as the target. To increase efficacy of these vaccines against SARS-CoV-2 variants, different strategies have been tested including administration of multiple vaccine doses and using spike protein with substitutions to match the latest variant in multiple subdomains as target antigen. Despite all these efforts, immunization rates are low in some populations due to limited supply or vaccine hesitancy. Consequently, protection against SARS-CoV-2 is poor resulting in locally evolving variants that impact the effectiveness of treatments. Kunze and colleagues showed that locally sourced convalescent plasma used for CPT in COVID-19 patients was associated with lower mortality than distantly sourced convalescent plasma, which suggested the immune responses show geographical differences [39]. 

As the humoral immune response between individuals becomes more heterogeneous due to emerging variants, different types of vaccines and vaccination rates, serological assays which only use the entire S and N proteins as antigen targets would not be sufficient to study the antibody repertoire of infected or vaccinated people. Protein microarrays developed with broad antigen targets such as N and S protein with its domains/subdomains, variants, and other human coronavirus antigens (cross reactivity) would be an ideal tool for determining detailed antibody signatures. Correlation, or otherwise, of these antibody signatures with clinical outcome would allow for optimization of future vaccines, and identification of potential leads for monoclonal antibody treatment.

2) Please, answer all the questions raised by the both reviewers.

Journal Requirements:

-We have corrected the manuscript and figure file names to meet PLOS ONE’s style requirements. 

-The original uncropped and unadjusted images for the SDS-PAGE gels are submitted as Supporting Information files.

3. Please upload a copy of Figure 5, to which you refer in your text on page 17. If the figure is no longer to be included as part of the submission please remove all reference to it within the text.

-This is corrected to Fig S4 in the revised manuscript.

Reviewers' comments:

Reviewer's Responses to Questions

Comments to the Author

1. Is the manuscript technically sound, and do the data support the conclusions?

Reviewer #1: Yes

Reviewer #2: Partly

2. Has the statistical analysis been performed appropriately and rigorously? 

Reviewer #1: Yes

Reviewer #2: N/A

3. Have the authors made all data underlying the findings in their manuscript fully available?

Reviewer #1: Yes

Reviewer #2: Yes

4. Is the manuscript presented in an intelligible fashion and written in standard English?

Reviewer #1: Yes

Reviewer #2: Yes

5. Review Comments to the Author

Reviewer #1: The manuscript "SARS-CoV-2 multi-antigen protein microarray for detailed characterization of antibody" by Alev Celikgil et al. introduces a valuable technique to screen individuals against multiple targets of SARS-CoV-2 antigens and variants in the same subarray with single small volume. The authors improved previously reported microarrays by using mammalian cell-derived antigens and showed the possibility of understanding the disease progress, detecting the efficacy of convalescent plasma transfusing therapy, or applying this technique to point-of-care testing.

Although this technique is quite interesting, some parts of this manuscript need to be explained for readers. Therefore, this reviewer would like to suggest some parts that should be clarified.

Major points:

1) The authors are encouraged to clearly state the rationale for developing this technique. As the authors mentioned in the Discussion, other microarrays against SARS-CoV-2 are already available. The advantageous ore incremental points for developing this technique should be explained in the Introduction compared to the others.

-Thank you for your suggestion. We have explained this on lines 71-77 of page 4 in the Introduction of the revised manuscript. 

2) The authors focused on the only 4 variants D614G, E484K, N501Y, and N501Y/Deletion69/70. How were these VOC selected with respect to any clinical relevance of these variants?

Spike variants were used in order to demonstrate the applicability of protein array technology in tracking responses to variant spike protein. In our protein array, we wanted to include mutations from each subdomain of S1 protein such as D614G (CTD), N501Y (RBD), and Deletion69/70 (NTD), which were found in the first SARS-CoV-2 variant with multiple substitutions (Alpha). The E484K substitution was important because it has shown to reduce susceptibility to single or combination antibody treatments in some studies. All three substitutions, D614G, E484K, and N501Y, were detected in multiple variants (Alpha, Beta, Gamma).

3) In Fig. 2A, according to the fitting curves of serum signal intensity against 6 concentrations of Spike antigen, the curve in 2.4 femto-mol was not likely to show linearity. Although the authors mentioned their protein array was compared with an ELISA at lines 232-239 of page 11 and Fig. S2, the authors are encouraged to show in detail the detection capacity of this technique compared to a conventional ELISA.

There was no difference observed in the detection capacity of the protein array compared to a conventional ELISA with the clinical samples and antigens tested. We showed this in S2 Fig in the original manuscript and explained on lines 253-255 of page 12 and 336-343 of page 16.

4) Table 1 indicates the information on blood samples the authors tested in this study. However, this reviewer could not follow which samples were used for which test sufficiently. For example, the authors mentioned, "Eleven of these patients were also evaluated for the antibody responses to S variants" in lines 88-89 of pages 4-5. Still, its result in Fig. 4B showed "Convalescent plasma samples" at the Y axis. This expression seemed to lead to misunderstanding whether the sample is from patients or donors.

This is corrected in Fig 2C and 4B, also clarified in the Table 1, the materials and methods and the results section of the revised manuscript.

5) On page 17, lines 354-356, it states, “Target and isotype differences of serum antibodies combined with clinical features may be useful predictors of disease progress for individuals infected with SARS-CoV-2." Even though the samples used in this study include a wide range of disease states, from severe to asymptomatic, no predictive analysis was performed. The authors should only mention this if there is confirmation of an ability to predict an individual outcome with findings using this technique.

This is corrected in the revised manuscript to ‘Target and isotype differences of serum antibodies is important for the immune assessment of individuals infected with SARS-CoV-2.’ (Line 376 on page 18).

6) Functional aspects of antibodies that contribute to protection from infection include not only antibody titer but also neutralizing activity and avidity. Of course, the microarray is intended to detect antibody titers. Therefore, the authors are required to explain the limitations of this technique.

Thank you for raising this important point; to address this we have added the following paragraph on pages 20-21 in the revised manuscript.

‘Although we used the SARS-CoV-2 protein array for detailed characterization of individuals’ immune responses, this study is limited to showing the antibody repertoire but not the efficacy of the antibody response in controlling infection. The protein array does not assess neutralizing activity, nor does it measure antibody avidity.’

Minor points:

1) The authors suddenly used the abbreviation "CPT" on line 128 of page 6 and line 170 of page 8. It may indicate "Convalescent Plasma Transfusing" or "Convalescent Plasma Therapy," the author should clarify the abbreviation.

This is clarified in the first instance on line 66 of page 3 in the revised manuscript.

2) "Fig. 5" at line 362 of page 17 may be incorrect. It seems "Fig. S4".

Thank you for the correction! This is corrected to Fig S4 in the revised manuscript.

Reviewer #2: The present study aims to design and produce immunogenic proteins of SARS-CoV2 (S and N) and different variants of S proteins and evaluate reactivity of serum sample of infected patients on a multi-antigen protein microarray.

Major issues:

-How such microarray could help as a POC or predict the outcome of the illness in patients according to the fact that the humoral immune response is very heterogenous in patients (as authors and previous similar works did not get any rational correlation between pattern of humoral immune responses and outcome of the disease). Moreover, the heterogeneity of humoral immune response become more and more complex as people infected with new emerging variants and more than 80% of them have vaccinated with different kinds vaccines eg. Inactivated whole virus, S, S1 and RBD subunit vaccine.

- Thank you for raising this important point; precisely because of the heterogeneity of humoral immune response between individuals, studying serum responses with serological assays that only use the full-length S and N proteins as antigen targets would not be sufficient for immune assessment. Microarrays developed with broad antigen targets such as N and S protein with domains/subdomains, variants, and other human coronavirus antigens (cross reactivity) would be an ideal tool for detailed antibody signatures and diagnostic purposes as POC, if not enough to predict disease outcome alone. We discussed this further on lines 413-439 on pages 19-21 of the discussion section in the revised manuscript. 

-Moreover, since the beginning of pandemic, medical/research institutes have been collecting and storing sera from SARS-CoV-2 infected and vaccinated individuals along with their clinical history and disease/vaccine timeline. Future studies extended to large number of clinical samples using multi-antigen protein arrays for serological assays in parallel with neutralization assays may be useful to better understand protective efficacy of individuals’ antibodies and disease progress.

-How is it possible after normalization, specific IgG against RBD is higher than specific IgG against S1 in keeping with the truth that the molar ratio of coating RBD and S1 recombinant protein are equal. In that case, S1 should have more binding epitopes than RBD!

-Lower S1 specific IgG antibodies than RBD specific IgG antibodies may be observed because non-RBD specific antibodies sterically hinder the access of RBD specific antibodies to RBD domain when whole S1 domain is used as the target antigen.

-Alternatively, the apparent lower level of S1 specific IgG antibodies than RBD specific IgG because may be antibody binding to S1 or RBD domain is not at saturation levels. In Fig 3, we are showing the data only from single dilution and single antigen concentration as an example of observed differences between serum samples (1/270 dilution, 7.2 fmol antigen concentration, with His Normalization). Following sentence is added to the Fig 3 legend for clarification in the revised manuscript. 

‘The data is shown for a single dilution (1/270) and 7.2 fmol antigen concentration.’

-Although the SDS-PAGE and anti-His tag antibody indicating presence of the HR domain, there is not any sign of reactivity of serum samples with this region, even positive control is not working!

The anti-His tag antibody is used to demonstrate that target antigens are immobilized on the slide. The positive control we used for serum screening is a convalescent plasma sample with known high Spike antibody levels (determined with ELISA by our collaborators) but not tested for any other antigen specific antibodies. As the positive control is a convalescent plasma sample, and not recombinant protein, it did not show a response to the anti-His antibody. This has been clarified in S2 Figure legend in the revised manuscript. We did not detect any antibodies against HR domain in any of the sera which suggests that it is not an immunogenic target for SARS-CoV-2. Again, this shows that multi-antigen protein arrays could be a useful tool for discovering immunogenic (or not-immunogenic) target antigens for effective therapies against SARS-CoV-2.

-The reactivity pattern of serum belongs to the 12 recovered convalescent donors, which mentioned in abstract and table1, was to properly determined in the MS.

This is clarified in the materials/methods and the results section of the revised manuscript, and Table 1.

-Different studies have shown that there are cross-reactive antibodies against N and S proteins in serum patient with other coronavirus family which could affect the interpretation of the results, why the authors have not used such controls in their experiments.

SARS-Cov-2 S protein shares the highest protein similarity with SARS-CoV S protein compared to other human coronaviruses. It was also shown previously that antibodies against SARS-CoV-2 N protein only cross-react with SARS-CoV N protein*. RBD domain of SARS-CoV was included in the protein array as shown in Figure 1A, C, D and mentioned in the materials/methods section of the manuscript. While the sera we have tested in this study was positive for SARS-CoV-2 N protein, it did not react with SARS-CoV RBD protein. 

As we intended to focus on spike subdomains and variants, this was not discussed in the manuscript before. In the revised manuscript we have mentioned this on pages 10 and 13 of the results section, and on pages 16-17 of the discussion.

‘RBD domain of SARS-CoV was also included for its cross-reactivity with SARS-CoV-2.’

‘There was no antibody response to RBD domain of SARS-CoV in any of the serum samples in our screening (representative image is shown in Fig 1D).’

‘RBD of SARS-CoV was also included in the array for cross-reactivity due to high protein similarity between SARS-CoV and SAR-CoV-2.’

‘SARS-Cov-2 S protein shares the highest protein similarity with SARS-CoV S protein compared to other human coronaviruses. Furthermore, Du and colleagues who developed a protein array using only S and N proteins from SARS-CoV-2 and other human coronaviruses showed that antibodies against SARS-CoV-2 N protein cross-reacts only with SARS-CoV N protein. While all the individuals we have tested in this study was positive for SARS-CoV-2 N protein, they did not show any antibody response to RBD domain of SARS-CoV.’

*We have added the following reference to the revised manuscript.

‘Development and Application of Human Coronavirus Protein Microarray for Specificity Analysis. Du et al 2021, Anal. Chem. 2021 Jun 1;93(21):7690-7698.’

Minor Issues:

-Abbreviation of “huAche” should be added at page 7 line 148

This abbreviation is added to the revised manuscript (line 164).

-Using day-1 and day1 in figure1 and description in result section is unclear.

This is clarified in the figure 1D, figure legend and in the result section on lines 222-223 of page 11 in the revised manuscript.

- The origin of expressed N protein at page 9 line 186 is unclear.

‘SARS-CoV-2’ is added for clarification on page 10 line 200 in the revised manuscript.

- The concentration of antigens has not mentioned in result section at page 11 lines 226-230.

It is added in the Fig 2 legend on page 12 in the results section of the revised manuscript.

- Why positive control in S2 figure has not any reactivity with HR antigen.

Please see the explanation in major issues section above.

- Where is the figure 5 which has been mentioned at page 17 line 362!

It is corrected to Fig S4 on page 18 line 382 in the revised manuscript.

6. PLOS authors have the option to publish the peer review history of their article (what does this mean?). If published, this will include your full peer review and any attached files.

Do you want your identity to be public for this peer review? For information about this choice, including consent withdrawal, please see our Privacy Policy.

Reviewer #1: No

Reviewer #2: No

---

## [Decision Letter · Decision Letter 1]

16 Jan 2023

SARS-CoV-2 multi-antigen protein microarray for detailed characterization of antibody responses in COVID-19 patients

PONE-D-22-28228R1

Dear Dr. Celikgil,

We’re pleased to inform you that your manuscript has been judged scientifically suitable for publication and will be formally accepted for publication once it meets all outstanding technical requirements.

Kind regards,

Paulo Lee Ho, Ph.D.

Academic Editor

PLOS ONE

Additional Editor Comments (optional):

Reviewers' comments:

Reviewer's Responses to Questions

**Comments to the Author**

1. If the authors have adequately addressed your comments raised in a previous round of review and you feel that this manuscript is now acceptable for publication, you may indicate that here to bypass the “Comments to the Author” section, enter your conflict of interest statement in the “Confidential to Editor” section, and submit your "Accept" recommendation.

Reviewer #1: All comments have been addressed

Reviewer #2: All comments have been addressed

2. Is the manuscript technically sound, and do the data support the conclusions?

Reviewer #1: Yes

Reviewer #2: Yes

3. Has the statistical analysis been performed appropriately and rigorously? 

Reviewer #1: Yes

Reviewer #2: Yes

4. Have the authors made all data underlying the findings in their manuscript fully available?

Reviewer #1: Yes

Reviewer #2: Yes

5. Is the manuscript presented in an intelligible fashion and written in standard English?

Reviewer #1: Yes

Reviewer #2: Yes

6. Review Comments to the Author

Reviewer #1: I am grateful to the authors for considering the suggested points, which were almost completely included.

Reviewer #2: (No Response)

7. PLOS authors have the option to publish the peer review history of their article (what does this mean?). If published, this will include your full peer review and any attached files.

Reviewer #1: No

Reviewer #2: No

---

## [Editor Report · Acceptance letter]

31 Jan 2023

PONE-D-22-28228R1 

SARS-CoV-2 multi-antigen protein microarray for detailed characterization of antibody responses in COVID-19 patients 

Dear Dr. Celikgil:

I'm pleased to inform you that your manuscript has been deemed suitable for publication in PLOS ONE. Congratulations! Your manuscript is now with our production department. 

Kind regards, 

on behalf of

Dr. Paulo Lee Ho 

Academic Editor

PLOS ONE